# Algorithm for diagnosis of early *Schistosoma haematobium* using prodromal signs and symptoms in pre-school age children in an endemic district in Zimbabwe

Tariro L. Mduluza-Jokonya[1]*, Arthur Vengesai[1], Herald Midzi[1,2], Maritha Kasambala[2], Luxwell Jokonya[1,3], Thajasvarie Naicker[1], Takafira Mduluza[1,2]

1 Optics & Imaging, Doris Duke Medical Research Institute, College of Health Sciences, University of KwaZulu-Natal; KwaZulu-Natal, Durban, South Africa, 2 Department of Biochemistry, University of Zimbabwe, Harare, Zimbabwe, 3 Department of Surgery, College of Health Sciences, University of Zimbabwe, Harare, Zimbabwe

* tljokonya@gmail.com

## Abstract

### Introduction

Prompt diagnosis of acute schistosomiasis benefits the individual and provides opportunities for early public health intervention. In endemic areas schistosomiasis is usually contracted during the first 5 years of life, thus it is critical to look at how the infection manifests in this age group. The aim of this study was to describe the prodromal signs and symptoms of early schistosomiasis infection, correlate these with early disease progression and risk score to develop an easy to use clinical algorithm to identify early *Schistosoma haematobium* infection cases in resource limited settings.

### Methodology

Two hundred and four, preschool age children who were lifelong residence of a schistosomiasis endemic district and at high risk of acquiring schistosomiasis were followed up from July 2019 to December 2019, during high transmission season. The children received interval and standard full clinical evaluations and laboratory investigations for schistosomiasis by clinicians blinded from their schistosomiasis infection status. Diagnosis of *S. haematobium* was by urine filtration collected over three consecutive days. Signs and symptoms of schistosomiasis at first examination visit were compared to follow-up visits. Signs and symptoms common on the last schistosomiasis negative visit (before a subsequent positive) were assigned as early schistosomiasis infection (ESI), after possible alternative causes were ruled out. Logistic regression identified clinical predictors. A model based score was assigned to each predictor to create a risk for every child. An algorithm was created based on the predictor risk scores and validated on a separate cohort of 537 preschool age children.

**Data Availability Statement:** The data underlying the results presented in the study are available

from the University of Zimbabwe College of Health Sciences Research Support Centre: Precious Chandiwana: precious.chandiwana@gmail.com and rsc@medschool.ac.zw http://uzchsrsc.ac.zw.

**Funding:** The study received funding from the TIBA to TIBA-Zimbabwe (TM). This research was commissioned by the National Institute of Health Research (NIHR), Global Health Research Programme (16/136/33) using UK aid from UK Government. The views expressed in this publication are those of the authors and not necessarily those of the NIHR or the Department of Health and Social Care. The funders had no role in study design, data collection and analysis, decision to publish, or preparation of the manuscript.

**Competing interests:** The authors have declared that no competing interests exist.

## Results

Twenty-one percent (42) of the participants were negative for *S. haematobium* infection at baseline but turned positive at follow-up. The ESI participants at the preceding *S. haematobium* negative visit had the following prodromal signs and symptoms in comparison to non-ESI participants; pruritic rash adjusted odds ratio (AOR) = 21.52 (95% CI 6.38–72.66), fever AOR = 82 (95% CI 10.98–612), abdominal pain AOR = 2.6 (95% CI 1.25–5.43), pallor AOR = 4 (95% CI 1.44–11.12) and a history of facial/body swelling within the previous month AOR = 7.31 (95% CI 3.49–15.33). Furthermore 16% of the ESI group had mild normocytic anaemia, whilst 2% had moderate normocytic anaemia. A risk score model was created using a rounded integer from the relative risks ratios. The diagnostic algorithm created had a sensitivity of 81% and a specificity of 96.9%, Positive predictive value = 87.2% and NPV was 95.2%. The area under the curve for the algorithm was 0.93 (0.90–0.97) in comparison with the urine dipstick AUC = 0.58 (0.48–0.69). There was a similar appearance in the validation cohort as in the derivative cohort.

## Conclusion

This study demonstrates for the first time prodromal signs and symptoms associated with early *S. haematobium* infection in pre-school age children. These prodromal signs and symptoms pave way for early intervention and management, thus decreasing the harm of late diagnosis. Our algorithm has the potential to assist in risk-stratifying pre-school age children for early *S. haematobium* infection. Independent validation of the algorithm on another cohort is needed to assess the utility further.

## Author summary

*Schistosoma haematobium* causes urogenital infection and in endemic areas schistosomiasis is usually contracted during the first 5 years of life, thus it is critical to look at how the infection manifests in this age group. Prompt diagnosis of acute schistosomiasis is required to benefit the individuals and provide opportunities for early treatment and public health intervention. The study examined symptoms that correlated with early disease progression and risk scored to develop an easy to use clinical algorithm to identify early *S. haematobium* infection cases. The children received standard full clinical evaluations by clinicians who were blinded from schistosomiasis diagnosis by parasitological examination. An algorithm was created based on the predictor risk scores and participants had the following prodromal signs and symptoms; pruritic rash, abdominal pain, pallor, abdominal pain, inguinal lymphadenopathy and a history of facial/body swelling within the previous month. A risk score model, diagnostic algorithm, was created that compared to urine dipstick and parasitology. This study demonstrates the clinical signs and symptoms associated with early *S. haematobium* infection in pre-school age children. These prodromal signs and symptoms pave way for early intervention and management, thus decreasing the harm of late diagnosis common in populations from endemic areas.

## Introduction

*Schistosoma haematobium* (urinary blood fluke) is a trematode species which belongs to the genus *Schistosoma* [1]. It is the second most prevalent parasitic infection in humans, found in Africa and the Middle East. *S. haematobium* is the major agent of schistosomiasis infection [2]. According to WHO estimates, 200 000 people per annum die from schistosomiasis [3]. In Zimbabwe there are two common *Schistosoma* species, *Schistosoma haematobium* and *Schistosoma mansoni*, with the former being more common in the country [4].

*S. haematobium* infection has an early, acute and chronic phase [5]. It is recorded that the early signs and symptoms are less pronounced, and in endemic regions they don't recur in the case of reinfection. In the acute phase, *S. haematobium* infection manifest as Katayama fever [6], that occurs from 4 weeks post infection and the symptoms include chills, fever, headache, nausea, vomiting, diarrhoea, dry cough, hepato-splenomegaly and lymphadenopathy [7]. Katayama fever is mostly noted in *Schistosoma* naïve travellers. The chronic phase of *S. haematobium* infection begin 12–16 weeks post infection [4].

*S. haematobium* infection can be diagnosed via direct and indirect diagnostic tests [8]. Direct diagnostic tests are parasitological diagnosis (egg detection via microscopy) from the urine or from organ samples in clinical settings [9]. This is the most preferred diagnostic method in surveys [10]. There are a number of indirect diagnostic tests, which are: immunological diagnosis (intradermal test and antibody detection), DNA and RNA detection-based methods or cytokines metabolic products and other parasite molecule biomarkers [11]. Though these tests are more advanced and have better specificity, they are not readily available in most schistosomiasis endemic areas due to costs.

The most widely used diagnostic test in limited resource setting, as in our case, is direct parasitology diagnosis via urine filtration and concentration of *S. haematobium* eggs, followed by microscopic examination [8]. Or in the case of clinical setting direct diagnosis can be made from biopsy [12]. However, none of the available direct egg detection tests are appropriate for early schistosomiasis diagnosis prior to parasite becoming patent. This is due to the fact that oviposition in urinary schistosomiasis commences around 4–6 weeks post cercarial infection [13].

*S. haematobium* infection morbidity in the chronic phase is devastating, thus it is imperative to halt the progression during the early phase. Early phase signs and symptoms can be referred to as prodromal signs. Prodromal signs and symptoms, are early indicators of an illness, which manifest before the disease can test positive on laboratory testing [14]. Prodromal signs and symptoms give a chance for early intervention and management in the course of a disease before it is fully blown. In the early phase of schistosomiasis urine filtration test is negative [12]. *S. haematobium* can be cured without progressing to complications with accurate diagnosis and prompt treatment during the initial stages of infection. Hence, the use of appropriate, sensitive diagnostic tools to identify infected individuals becomes imperative especially in preschool age children (PSAC) in endemic areas.

To our knowledge, no studies have been done to look at possible prodromal signs and symptoms in pre-school age children infested with S. *haematobium*. Though it is well known that first exposure of schistosomiasis occurs in this age group. We set out to investigate *S. haematobium* infection associated prodromal signs and symptoms in PSAC from an endemic area in order to create a viable algorithm that can be used in resource limited settings.

## Methodology

### Ethics statement

Ethical approval was obtained from Medical Research Council of Zimbabwe (MRCZ/B/1854). Approval was also obtained from the Provincial and District Medical Directors and

Community Leaders. Written informed consent was obtained from the parent/guardian of the children. Children diagnosed of other diseases were managed appropriately and according to W.H.O standards. For confidentiality purposes all participants were allocated a number and underwent individual counselling and private treatment depending on their schistosomiasis and other infection status.

## Study site and design

The study was carried out in Shamva district located 31˚40′0” E longitude and 17˚10′0” S latitude in Mashonaland Central province, Zimbabwe [15]. The area lies 945 m above sea level, with a warm and temperate climate average temperature being 20.2˚C and annual rainfall of 887 mm. Located in the Mazowe valley, Shamva district is located in an area with high farming activity due to its fertile soil and abundant rivers for water supply. Residents get their water supply from Mupfure and Mazowe rivers and their tributaries which are a huge source of alluvial gold. The vast majority of the population survives through gold panning [15]. Shamva district has the highest prevalence of schistosomiasis in the Zimbabwe at 62.3% [16]. The study was a longitudinal study done between July 2019 to December 2019 a period where there is the highest schistosomiasis transmission during the hottest months (September–December) [17]. A cohort of 314 preschool age children at high risk of schistosomiasis infection were screened. The first visit was in July 2019 and 204 *S. haematobium* negative but high risk PSAC were recruited for follow-up, the second visit was four weeks later and the third visit was 16 weeks later. We chose these time lines as this is the period it takes between *S. haematobium* infestation and establishment of infection seen as haematuria or ova in urine [18,19]. After creation of the algorithm we selected a validation cohort which was from the same endemic district but from different villages with a total of 537 PSAC. Parents and guardians of participants underwent risk reduction counselling at each visit.

## Study inclusion criteria

The participants recruited into the study were children under the age of 5 years, who met following inclusion criteria: 1. Have parental/guardian consent to participate, 2. Be lifelong residents of the study area, 3. Have no previous anti-helminthic treatment exposure, 4. Be negative for *Schistosoma mansoni* and geohelminths, 5. Be negative for the ToRCHeS (toxoplasmosis, rubella, cytomegalovirus, hepatitis and syphilis) screen, 6. Had a widal TO ratio <1:160, 7. Have no malarial infection, 8. Be HIV negative and have no exposure to HIV, 9. Had no other illness manifesting (between follow-ups), 10. Mantoux test reaction <5mm and 11. Have a normal nutrition status based on clinical examination, mid upper arm circumference measurement and weight-for-age as well as height-for-age measurements. The children observed to have active infection were managed appropriately and referred to the District hospital if there was any need.

## Sample size

Participant selection was by simple random sampling. Mothers were requested to bring their children to the clinic or EPI (Expanded Program of Immunization) meeting points. The required sample size was calculated to be 113 participants using the formula as follows, were the known schistosomiasis prevalence in PSAC was 8.4% (4):

$$n = \frac{z^2 pq}{e^2}$$

Where z is the value for the 95% confidence interval, that is alpha of 5% ($z = 1.96$)

p = proportion/prevalence of the outcome to be investigated ($p = 0.08$)

q = 1-$p$ = 0.92

e = precision for the given confidence interval expected as a decimal ($e = 0.05$)

$n = 113$.

## Clinical examinations

Children were evaluated at baseline and on follow-ups using clinical examination protocol according to standard clinical examinations process [20,21]; and they were treated accordingly. A questionnaire was administered to caregivers by the examining clinicians as part of history taking to screen for risk factors and symptoms associated with the prodromal phase of *S. haematobium* infection. The clinical examinations were conducted by three medical practitioners who were blinded to the infection status of the participant to avoid bias. Diagnosis of prodromal phase of schistosomiasis was made in retrospect on a child who tested negative for schistosomiasis and then eventually tested positive in the subsequent visit. All tests were done on 3 consecutive days during each visit. Parents and guardians of participants underwent risk reduction counselling on each visit.

## Microhaematuria screen

Urine samples collected were examined for microhaematuria using the Uristix reagent strips (Uripath, Plasmatec, UK) dipped into fresh, well-mixed urine for 40 sec and the test area was compared with a standard colour chart as per manufacturer's instructions.

## Parasitology diagnosis

Urine samples were collected in labeled wide mouth containers over at least two consecutive days for each individual because of daily variations in ova excretions in urine. Children under one year used pediatric urine collectors attached by a clinical worker. The samples were collected between 10am and 2pm for the older children. The caregivers then brought the samples which were examined as follows using the urine filtration technique as previously described [22].

Stool samples were collected on a single day and processed using the Kato-Katz method with 2 slides prepared per sample, parasite eggs were enumerated under a light microscope for *S. mansoni* in duplicate and results expressed as eggs per gram of stool. Formal ether sedimentation technique was used to look for geohelminths as previously described [23]. The parasitology examination was conducted by the parasitology team and results obtained recorded separately without access by the clinical team.

## Blood processing and analysis

Plasma and sera were obtained from blood collected in well-labeled EDTA and plain blood tubes, respectively. Full blood count and urea and electrolytes were processed. Anaemia was classified according to a World Health Organisation protocol on haemoglobin (Hb) concentrations for the diagnosis of anaemia and assessment of severity [24]. The scale we used for children 6–59 months was classified as follows: 1. Hb>11g/dl = non anaemic, 2. Hb 10–10.9g/dl = mild anaemia, 3. Hb 7–9.9g/dl = moderate anaemia and 4. Hb <7g/dl = severe anaemia.

Serum and plasma obtained from each child were processed and tested for common paediatric infections in Zimbabwe: Toxoplasmosis, rubella, cytomegalovirus, herpes simplex (ToRCHeS) virus 1 and 2, HIV and Hepatitis (A, B and C) using the Maglumi 4000

chemiluminescence immunoassay analyser (CLIA). Thick blood film slides were stained using the Giemsa stain and examined for malaria parasites using microscopy.

## Statistical method

Initially, we sought a relationship between the signs and symptoms labeled as prodromal and *S. haematobium* infection. Data analysis was performed using Stata 15. The statistical methods applied included the descriptive statistics, bivariate and multivariate logistic regression modelling.

The multivariate logistic regression models were fitted to adjust for potential confounding factors for the five manifestations with three explanatory variables, that is, sex, age and schistosomiasis infection. The effect of different factors on the prevalence of schistosome infection and morbidity was determined using logistic regression and the results reported as adjusted ORs (AORs) and 95% CI, along with the test for significance, as previously described [25]. Generalized Linear Model was used to prove the null hypothesis of the different factors using the P-values after factoring in age, sex and infection intensity. Secondly, we created a predictor risk score model using variables which were associated with early schistosomiasis infection at p< 0.05 which were then included in the final model, then used to develop a model-based score to predict early schistosomiasis infection. In addition to applying a risk score model to predict ESI, the absolute number of signs/symptoms was also assessed to determine association with ESI. Sensitivity, specificity and positive likelihood ratios (with 95% confidence intervals) for detecting ESI were calculated for different numbers of signs and symptoms. Infection status for *S. haematobium* was defined as the arithmetic mean egg count/10mL of at least two urine samples collected on three consecutive days. To create the algorithm, we estimated using the grey zone approach, the value for which the scores did not provide conclusive information. After acquiring definitive scores using the Youden's index coupled with the receiver operating characteristic and creating the risk predictor scores, we then assessed the performance of our scores and algorithm compared to the schistosomiasis screening gold standard (dipstick). Finally, we validated our findings in a separate cohort of 537 PSAC from an endemic area at high risk of *S. haematobium* infection.

## Results

### Demographics

A total of 204 children from 4 villages in Shamva district, Mashonaland central, Zimbabwe were followed up between July 2019 to December 2019 for the algorithm derivation cohort. At baseline we screened 314 children, 96 of which did not meet the inclusion criteria and 14 got lost to follow-up mostly due to relocation (**Fig 1**). In the algorithm derivation cohort (ADC) males were 52% (106/204) and the rest were females. The age was normally distributed with mean (SD) age of 3.39 (1.08) years (**Table 1**). For the algorithm validation cohort (AVC) a total of 537 PSAC met the inclusion criteria, 51% (272/537) were males.

### *S. haematobium* infection trends

*S. haematobium* prevalence at first follow-up was 20% (40/204) with no significant difference on gender (**Table 2**). At second follow-up the prevalence was 28% (57/204), 21% (42/204) had new infections after previous negative test reported as early schistosomiasis infection (ESI) and 32% (25/57) of the positives at first follow-up being negative following treatment. All participants who had ESI were haematuria negative. Most of the PSAC had low infection intensity

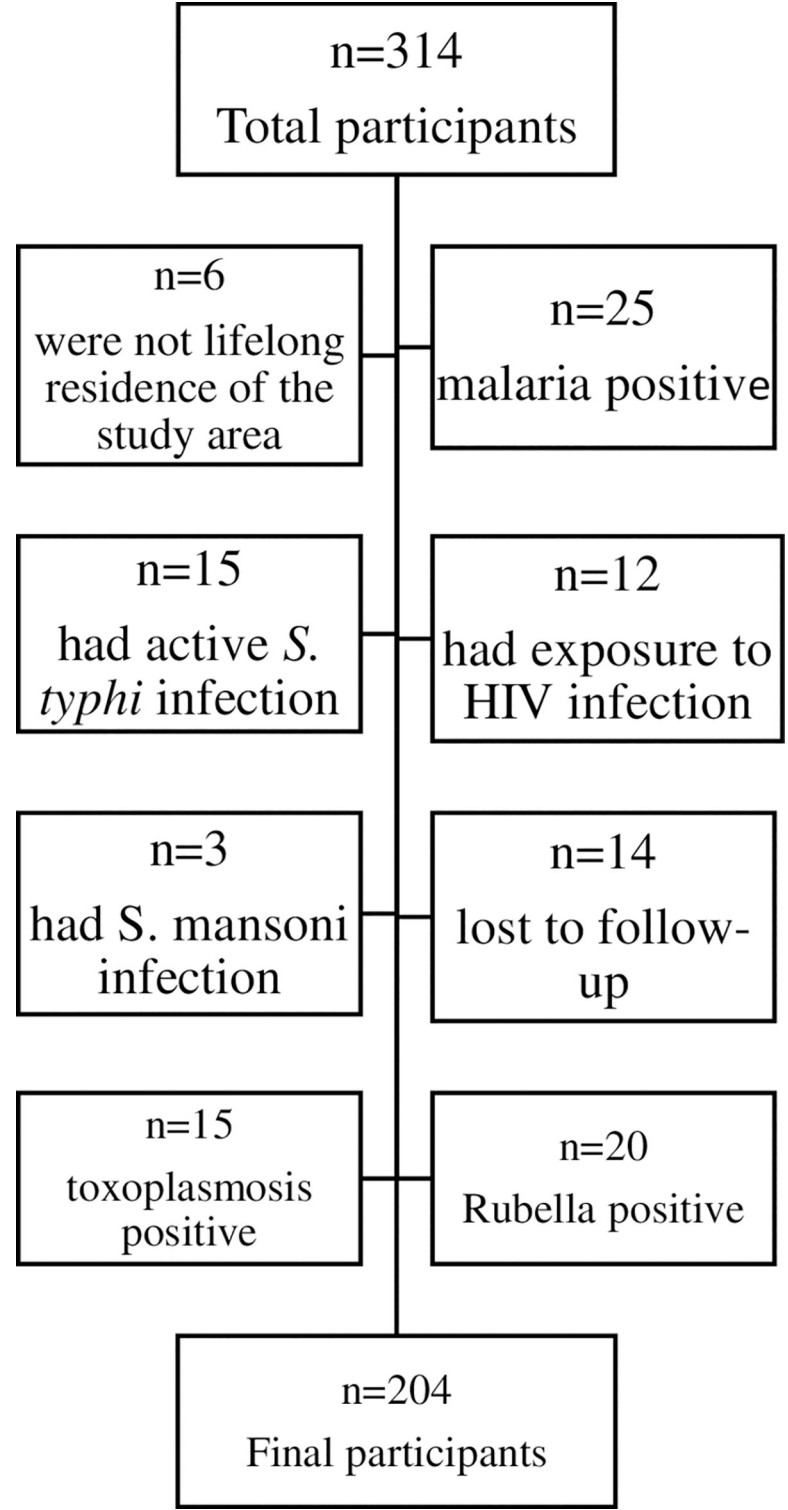

**Fig 1. The Flow Chart showing the infections encountered in the study population and Selecting the Final Study Participants.**

**Table 1. Characteristics of Study Participants.**

| Characteristic | | Algorithm Derivation Cohort | | | Algorithm Validation Cohort % (n) (N = 537) |
|---|---|---|---|---|---|
| | | *S. haematobium* negative %(n) (N = 162) | | Early schistosomiasis infected (ESI) participants %(n) (N = 42) | |
| | | @ 4 weeks | @16 weeks | | |
| Sex | Male | 52% (106) | 49% (80) | 24% (26) | 51% (272) |
| | Female | 48% (98) | 50% (82) | 16% (16) | 49% (263) |
| Family member with schistosomiasis/treated in the recent past | yes | 83% (169) | 79% (128) | 98% (41) | 92% (494) |
| | No | 17% (35) | 21% (34) | 2% (1) | 8% (40) |
| Exposure to contaminated water | Yes | 90% (183 | 88% (142) | 98% (41) | 87% (468) |
| | No | 10% (21) | 12% (20) | 2% (1) | 13% (68) |
| Had previous antihelminthic treatment | Yes | 6.4% (13) | 6.8% (11) | 4% (2) | 4% (22) |
| | No | 93% (191) | 93% (151) | 96% (40) | 96% (514) |

across all survey time points. Participants in the ESI group were mostly male (62%) with remaining being females.

## Blood electrolytes and full blood count (FBC) values

Urea and electrolytes (kidney function) were essentially normal in all participants. Of the study participants 91.7% had no anaemia, 6.9% had mild anaemia and 1.5% had moderate anaemia (Fig 2). ESI children were more likely to have mild normocytic anaemia in comparison to the negative participant, Relative risk (RR) = 3.86 (1.43–10.39) p = 0.007. None of the

**Table 2. Characteristics of study participants at follow-ups.**

| Variables | | First follow-up (*S. haematobium* infected participants) @ 4 weeks n (%) (N = 40) | Second follow-up (*S. haematobium* infected participants) @ 16 weeks n (%) (N = 57) | Early *S. haematobium* infection (ESI) participants n (%) (N = 42) |
|---|---|---|---|---|
| Sex | Male | 48% (19) | 56% (32) | 62% (26) |
| | Female | 52% (21) | 44% (25) | 38% (16) |
| Infection intensity | Low < 10 eggs/10mls of urine | 92% (37) | 98% (56) | 81% (34) |
| | Mild >10 to < 50 eggs/10mls of urine | 3% (1) | 0% | 12% (5) |
| | High >50 eggs/10mls of urine | 5% (2) | 2% (1) | 7% (3) |
| Haematuria | Negative | 33% (13) | 84% (48) | 100% (42) |
| | Positive | 67% (27) | 16% (9) | 0 |
| Schistosomiasis prevalence | | 20% (40) | 28% (57) | 21% (42) |

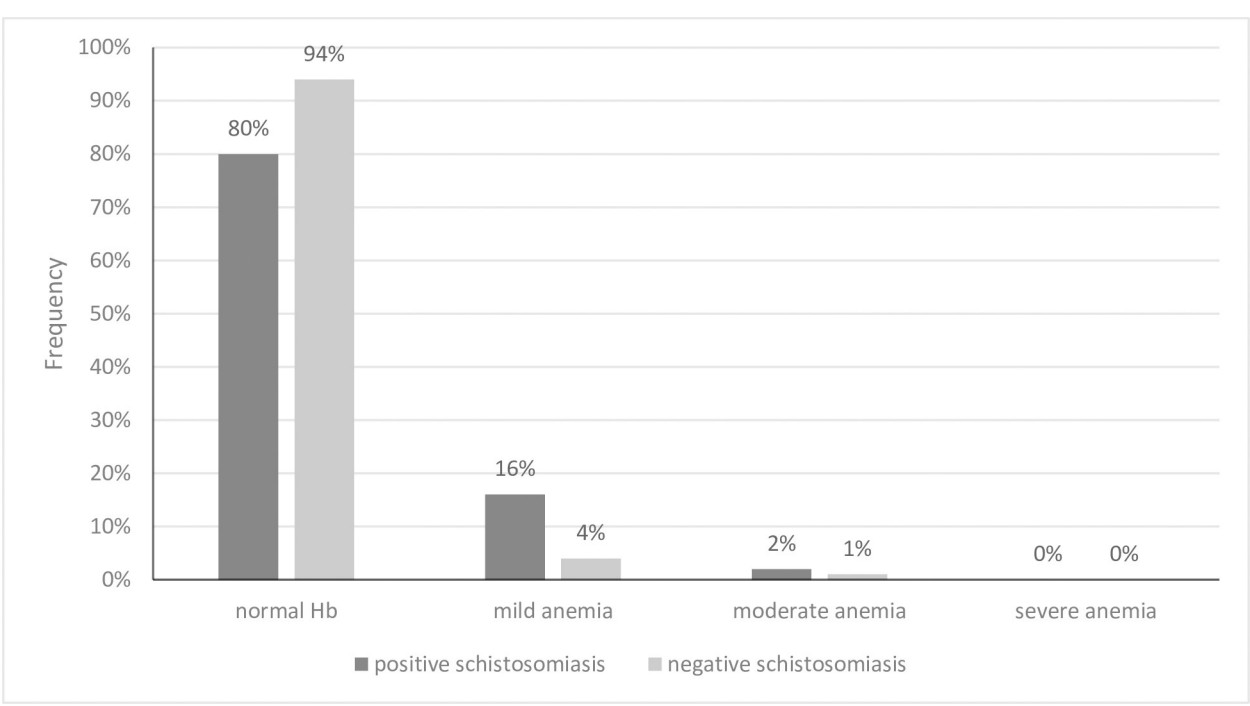

**Fig 2. Anaemia Prevalence Based on Haemoglobin Levels of the Study Participants.**

participants had severe anaemia and 2% of the ESI had moderate normocytic anaemia whilst 1% of the rest of participants had microcytic anaemia. Platelets were all within the normal ranges for the participant ages. White cell count (WCC) was mostly within the normal range with 1% of participants having a slightly raised value. WCC differential was unavailable.

### Prodromal *S. haematobium* signs and symptoms

ESI participants had the following signs and symptoms on their *S. haematobium* negative visits: pruritic rash (93%), fever (98%), abdominal pain (62%), pallor (81%), facial/body swelling within the month (64%), and inguinal lymphadenopathy 33% (**Table 3**). These prodromal signs and symptoms had the following adjusted odds ratio (AOR at 95% CI): pruritic rash AOR = 21.52 (95% CI 6.38–72.66), fever AOR = 82 (95% CI 10.98–612), abdominal pain AOR = 2.6 (95% CI 1.25–5.43), Pallor AOR = 4 (95% CI 1.44–11.12), atopy history AOR = 7.31 (95% CI 3.49–15.33) and inguinal lymphadenopathy AOR = 2.88 (95% CI 1.32–6.24).

All of the ESI participants reported at least one sign or symptom and the specificity and likelihood ratios increased with an increasing number of reported symptoms (**Table 4**). Participants with three signs and/or symptoms were more likely to have schistosomiasis infection at follow up compared to those with no signs/symptoms (AOR = 6.12: 95% CI: 1.08–34.79); p = 0.04); and presence of more than 4 signs/symptoms was strongly associated with ESI (AOR = 37.80; 95% CI: 6.2–44.7; p<0.001).

### Risk-score model predicting early *S. haematobium* infection

The variables significantly associated with the prodromal phase of schistosomiasis were used to construct a predictor risk score using the risk coefficients rounded off to the nearest integer. The risk score was calculated and shown in **Table 5** together with the predictor GLM p values.

**Table 3. New Infection of *S. haematobium* with Prodromal Signs and Symptoms at Baseline.**

| Predictors | Positives % (n) (N = 42) | Odds Ratio | Adjusted odds ratio (95% CI) | Relative risk (95% CI) | Sensitivity % | Specificity % |
|---|---|---|---|---|---|---|
| Risk factors | | | | | | |
| Family member with/had urogenital schistosomiasis | 98%(41) | 16.30(1.13–29.27)* | 10.89(1.44–82.05)* | 8.82(1.24–62.54)* | 24.3% | 97.1% |
| Frequent exposure to contaminated water | 96%(40) | 5.18(1.18–22.73)* | 4.92(1.13–21.45)* | 4.15(1.04–16.62)* | 23.5% | 94.12% |
| Signs and symptoms | | | | | | |
| Pruritic rash | 93% (39) | 22.96(6.20–70.91)* | 21.52 (6.38–72.66)** | 4.73(2.91–26.14)** | 92.9% | 62.3% |
| Fever | 98% (41) | 86.17(11.48–647)* | 82 (10.98–612.26)** | 8 (4.02–94.76)** | 97.6% | 66.7% |
| Abdominal pain | 76% (32) | 27.40(12.86–58.3)* | 13.52 (6.01–30.42)* | 1.89 (1.46–2.44) | 76.2% | 92.9% |
| Pallor | 81% (34) | 4.22(1.49–11.91)* | 64.6 (23.73–175.8)** | 4.18 (2.42–7.22)* | 81% | 93.8% |
| An episode of Facial/body swelling within the month | 64% (27) | 7.34(3.47–15.52)* | 7.31 (3.49–15.33)** | 3.25 (1.49–3.40)** | 64.3% | 80.2% |
| Inguinal lymphadenopathy | 33%(14) | 3.09(1.40–6.83)* | 2.88(1.32–6.24)** | 2.25(1.28–3.96) | 33.3% | 85.2% |

*significant at 5% level of significance

### The diagnostic algorithm

Using Youden's index determination for bootstrapped population and the alternative approach of splitting the receiver operating characteristic (ROC) into two curves of sensitivity and specificity, we retrieved an inconclusive zone spreading from 4–8 and 4.1–8.4, respectively and rounded the grey zone score to 4–8. The rate of confirmed schistosomiasis increased significantly from 0% (0/69) for score ≤3 to 59% (39/66) for score ≥9, 33% (68/204) participants were in the grey zone of 4–8 and confirmed cases were 0.04% (3/68).

Based on the results of the grey zone determination, we proposed a diagnostic algorithm in which the management includes the following: PSAC with a score of ≤3 would need only risk reduction counselling, a predictor score of 4–9 would overall need a risk sticker to alert healthcare workers to pay attention when the child visits next and a predictor score of ≥9 has a high probability of being *S. haematobium* infection (Fig 3). In resource-limited settings without readily available urine filtration facility and with shortage of dipsticks as in the case of the area where we did our study, we would strongly recommend for the child to be managed as a schistosomiasis patient.

The sensitivity and specificity of different cut-off scores were determined by comparing the risk score with the *S. haematobium* infection (Table 6). A score of ≤3 was less likely to predict *S. haematobium* infection with a sensitivity of 2.4% and a specificity of 58% whereas a score of ≥9 had a specificity of 96.9% and a sensitivity of 81%. Relative risk of the high risk group was 7.42 (95% CI 3.27–16.83; p = <0.001). The positive predictive value (PPV) was as follows for the different risk scores (RS): RS ≤3 (1.45%), RS 4–8 (16.3%) and RS ≥9 had a strong PPV of 87.2%. Negative Predictive Values (NPV) were as follows: RS ≤3 (69.6%), RS 4–8 (78.26%) and RS ≥9 (95.2%).

### External validation

There was a similar performance of the algorithm in the validation cohort as was in our initial cohort. The rate of confirmed schistosomiasis increased significantly from 0.1% (20/180) for

**Table 4. Sensitivity and Specificity of the Number of Signs and Symptoms.**

| Number of signs and symptoms | Sensitivity (%) | Specificity (%) | Relative Risk (95% CI) | Adjusted Odds Ratio (95% CI) |
|---|---|---|---|---|
| >1 | 100% | 32.7% | 1.39(1.26–1.52)* | - |
| >2 | 100% | 66.7% | 1.78 (1.49–2.12)* | - |
| >3 | 92.9% | 89.4% | 3.23 (2.17–4.80)* | 6.12 (1.08–34.79)* |
| >4 | 80.5% | 97.5% | 8.80 (3.48–22.22)* | 37.8 (6.2–44.70)* |

*significant at 5% level of significance

score ≤3 to 66% (94/142) for score ≥9, 43% (235/537) participants were in the grey zone of 4–8 and confirmed cases were 20% (48/235). Accuracy of the algorithm in the validation cohort is displayed in **Table 7**. The algorithm proposed by this study had a higher sensitivity and specificity in both the derivation (sensitivity = 81% and specificity = 96.9%) and validation cohort (sensitivity = 87.7% and specificity = 73.3%). Of interest, we showed that AUC increased markedly, from 0.58 using the standard screening tool (urine dipstick) to 0.93 when applying the algorithm.

## Discussion

The period when *S. haematobium* first infect an individual is crucial to note as chemotherapy can be used to stop transition to chronic schistosomiasis. This has the potential to decrease disability-adjusted life years (DALYs) associated with schistosomiasis in endemic areas. In this study we identify signs and symptoms predictive of the *S. haematobium* prodromal phase in PSAC in a schistosomiasis endemic area. We present a screening algorithm which can be used to enhance detection of *S. haematobium* infection in resource limited settings. Importantly, we report that rash, fever, abdominal pain, pallor, facial/body swelling with urticaria and inguinal lymphadenopathy were prodromal signs and symptoms for early *S. haematobium* infection. Furthermore we found an increase in prediction score with the number of symptoms which the child would have presented. A risk score criteria was created depending on the number of signs and symptom the child would have had at the *S. haematobium* infection negative visit, a risk score of ≥9 was highly associated with *S. haematobium* infection. Furthermore, the algorithm we created had better sensitivity and specificity compared to urine dipstick which is the common screening tool in resource limited settings.

The PSAC in the ESI group during the follow-up time point showed a strong association of having presented with a pruritic rash at the *S. haematobium* negative visit. This is in keeping with studies done on *Schistosoma* naïve travellers before they have Katayama fever [26,27] Transcutaneous penetration of the cercariae after contaminated water contact in the early phase may cause an allergic reaction manifesting as a pruritic maculopapular rash at the larvae

**Table 5. Risk Scores for the Different Factors and Predictor GLM.**

| Factor | Risk score | Predictor GLM, P Value |
|---|---|---|
| Fever | 4 | <0.0001 |
| Pallor | 2 | 0.005 |
| Inguinal lymphadenopathy | 1 | 0.002 |
| Frequent exposure to contaminated water | 1 | 0.013 |
| Family member complaining of haematuria or treated for schistosomiasis | 1 | 0.004 |
| Abdominal pain | 2 | 0.010 |
| An Episode of Facial/body swelling | 1 | <0.0001 |

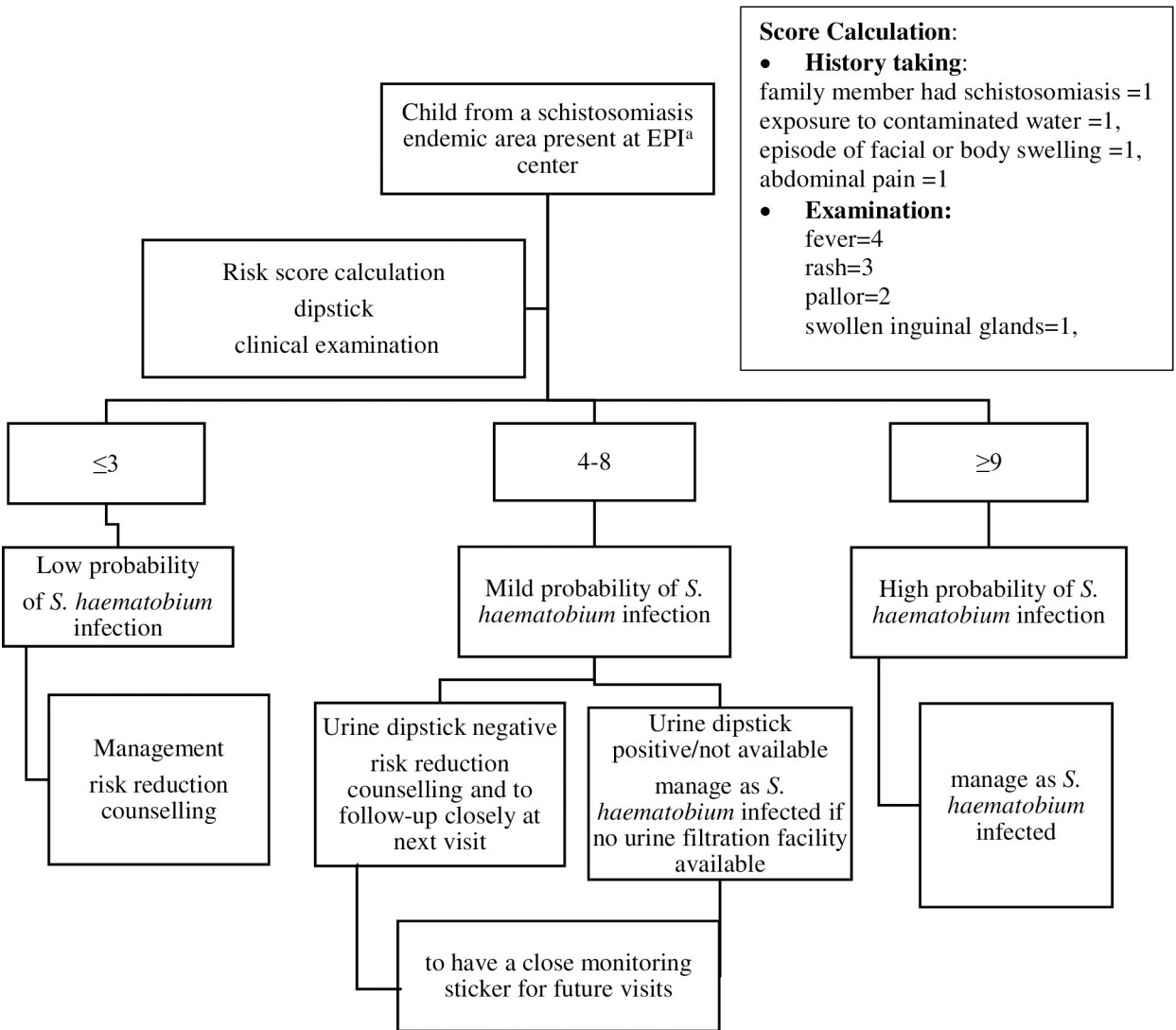

**Fig 3. The Algorithm for *S. haematobium* Diagnosis and Management for Early Schistosomiasis Infection in Preschool Age Children in Resource Limited Settings.**

penetration sites [13]. These signs are self-limiting in a few days. In the chronic phase, cutaneous lesions observed consists of granuloma clusters located in the dermis part of the skin [28]. Caregivers of the ESI participants also reported facial/body swelling with urticaria in their children at baseline. Other causes of swelling were eliminated and 75% of the children had no self

**Table 6. Risk Scores model.**

| Risk score | Definition | Sensitivity | Specificity | Risk (95%CI) | p-value | PPV [a] | NPV [b] |
|---|---|---|---|---|---|---|---|
| ≤3 | Low risk | 2.4% | 58% | **0.71(0.63–0.79)**[*] | < 0.0001 | 1.45% | 69.63% |
| 4–8 | Mild risk | 16.7% | 77.8% | 0.94 (0.80–1.09) | 0.433 | 16.3% | 78.26% |
| ≥9 | High risk | 81% | 96.9% | **7.42 (3.27–16.83)**[*] | <0.0001 | 87.2% | 95.2% |

[a] Positive Predictive Value, [b] Negative Predictive Value, [*]**at 5% level of significance**

**Table 7. Diagnostic performance of the algorithm versus urine dipstick in both the derivation and validation cohort.**

|  | Algorithm derivation cohort | Urine dipstick test in the algorithm derivation cohort | Algorithm validation cohort | Urine dipstick test in the algorithm validation cohort |
|---|---|---|---|---|
| AUC | 0.93 (0.90–0.97) | 0.58 (0.48–0.69) | 0.80 (0.74–0.83) | 0.66 (0.56–0.75) |
| Sensitivity | 81.0% | 0% | 87.7% | 17.9% |
| Specificity | 96.9% | 83.3% | 73.3% | 92.8% |
| Positive predictive value | 87.2% | 0% | 66.3% | 51.8% |
| Negative predictive value | 95.2% | 76.2% | 93.2% | 72.3% |

or family history of atopy. Furthermore their (renal function) urea and electrolytes at baseline and follow-up were normal. The caregivers considered the swelling and urticaria as nothing to worry about as most of the other older children also had episodes of similar presentation which self-resolved. An allergic reaction is reported to be triggered when cercariae penetrate into the skin [29]. This might be the reason behind this symptoms, however more research into this subject is warranted. In PSAC with low *S. haematobium* infection intensity, this is a valuable key observation that leads to a high index of suspicion and offers early diagnosis and treatment option, which can also be utilised by informed parents and community health workers.

Fever had a PPV of 43% and NPV of 99% for schistosomiasis positivity on follow-up. It had an accuracy of 73% which makes the symptom very relevant as an early *S. haematobium* infection marker. Schistosomiasis also known as snail fever has been noted to cause a fever in schistosomiasis naïve individuals [30]. Though it has not been noted in people from endemic regions, in this study we propose a link as PSAC are first exposed to *S. haematobium* infection similar to this phase [31]. Thus the immunological reaction triggered by infection could cause a fever in this fragile age group at their initial life course infection contact. Though fever is a symptom of numerous illnesses [28,29,32], the episodes reported here were not accompanied by acute symptoms of any other illness and were self-resolving.

Abdominal pain and pallor had PPV of 51% and 77% respectively. Abdominal morbidity is noted in schistosomiasis naïve travellers. In *S. mansoni* the ova triggers abdominal pain by being the antigenic stimulus producing an inflammatory reaction [33]. Schistosomiasis is associated with anaemia of inflammation noted to be due to the chronic inflammation observed in *S. haematobium* infection [34,35]. Since our participants live in a high schistosomiasis risk area we assume that they could have been exposed to chronic inflammation in-utero thus explaining the anaemia [31]. There is need to pursue the pathogenesis of these signs and symptom in-order to understand the manifestation of schistosomiasis in PSAC, useful towards syndromic diagnosis by parents/guardians and community health workers for early diagnosis.

One of the main reasons why PSAC are neglected in national schistosomiasis control program is because of the difficulties in acquiring a parasitology sample (especially in infants still in diapers) [36], we propose to curb this point by creating an easy-to-use the diagnostic algorithm we created. The diagnostic algorithm is composed of the prodromal signs we are reporting, which are non-specific so we strengthened the algorithm by adding *S. haematobium* infection risk factors which are more specific. We compared the diagnostic algorithm to urine dipstick, which is recommended for schistosomiasis screening. Our algorithm had better sensitivity, specificity, positive predictive value, negative predictive value in both the algorithm derived and the validation cohort compared to urine dipstick. The urine dipstick sensitivity, specificity, positive predictive value and negative predictive value was in keeping with other studies report [37,38].

This study demonstrates that it is possible to create a model predictive for *S. haematobium* infection in PSAC in endemic areas. This algorithm can be used to group PSAC into low, mild and high risk, and allow close monitoring of *S. haematobium* infection. In this study the risk algorithm created had a strong positive predictive value, this makes it a useful tool in screening of schistosomiasis before ova can be tested in urine in resource limited countries.

Praziquantel stage-specific susceptibility include the cercariae, very young schistosomula and adult worms thus giving PSAC a dose in the prodromal period has the potential to greatly reduce morbidity [39]. The strength of our study lies in that the study was a longitudinal cohort study, composed of PSAC from a high schistosomiasis prevalence area. The design included clinical assessment at each visit which allowed comparison of signs and symptoms, given possible concurrent of other illnesses, this assessment was an advantage to the study. The risk algorithm created had a strong positive predictive value, this makes it a useful tool in screening of schistosomiasis before ova can be tested in urine in resource limited countries.

Study limitation is we did not test using advanced laboratory diagnostic tools for schistosomiasis infection with the ability of earlier detection of schistosomiasis, we tried to correct this by validating the algorithm on a different larger cohort. Further, independent validation of the model in different settings would be necessary. Controversy might arise from the fact that the prodromal signs and symptoms we report are non-specific, we tried to counter this by adding *S. haematobium* infection specific risk factors to the algorithm. Possible differential diagnoses of the prodromal signs and symptoms were evaluated by use of relevant tests which were done thoroughly at the discretion of the attending doctors. In a setting where nucleic acid testing is not yet readily available, the ability to develop a checklist of signs/symptoms that would increase clinical suspicion of *S. haematobium* infection would be of benefit for triaging of high-risk PSAC and may lead to earlier detection and treatment of infection.

## Conclusion

In order to reach the goal of schistosomiasis elimination, we need an effective, inexpensive, easy to use and practical diagnostic test which can be used at a point-of-care facilities as well as at a community levels. Thus we propose our diagnostic algorithm for early schistosomiasis screening in PSAC in endemic low-resource setting areas. Our diagnostic algorithm has better performance compared to the standard screening tools. Furthermore, this study demonstrated prodromal signs and symptoms associated with early *S. haematobium* infection in PSAC. Using these prodromal signs and symptoms as early markers of infection will have a huge impact in early treatment which means the disease process will be halted before reaching the more harmful chronic stage of infection. We call upon the scientific community to look deeper into fever, localised pruritis, lower abdominal pain, a history of facial swelling and inguinal lymphadenopathy as prodromal markers of *S. haematobium* infection in PSAC, and for policy makers to consider our algorithm as a screening for children in endemic areas when they attend the routine clinic visits for growth and health monitoring.

## Acknowledgments

We would like to acknowledge the Ministry of Health and Child Care, the Medical Research Council of Zimbabwe, village health workers, nursing staff, parents and children from Shamva. A special thanks to members of the Biochemistry Department at the University of Zimbabwe for technical support during field parasitology and sampling. Our most profound gratitude to the participants and their parents or guardians for taking part in this study. The views expressed in this publication are those of the authors and not necessarily those of the NIHR or the Department of Health and Social Care.

## Author Contributions

**Conceptualization:** Tariro L. Mduluza-Jokonya, Thajasvarie Naicker, Takafira Mduluza.

**Data curation:** Tariro L. Mduluza-Jokonya, Takafira Mduluza.

**Formal analysis:** Tariro L. Mduluza-Jokonya, Arthur Vengesai, Herald Midzi, Maritha Kasambala, Luxwell Jokonya, Thajasvarie Naicker, Takafira Mduluza.

**Funding acquisition:** Takafira Mduluza.

**Investigation:** Tariro L. Mduluza-Jokonya, Arthur Vengesai, Herald Midzi, Maritha Kasambala, Luxwell Jokonya, Thajasvarie Naicker, Takafira Mduluza.

**Methodology:** Tariro L. Mduluza-Jokonya, Arthur Vengesai, Herald Midzi, Maritha Kasambala, Luxwell Jokonya, Thajasvarie Naicker, Takafira Mduluza.

**Project administration:** Tariro L. Mduluza-Jokonya, Takafira Mduluza.

**Resources:** Tariro L. Mduluza-Jokonya, Luxwell Jokonya, Thajasvarie Naicker, Takafira Mduluza.

**Software:** Tariro L. Mduluza-Jokonya.

**Supervision:** Thajasvarie Naicker, Takafira Mduluza.

**Validation:** Tariro L. Mduluza-Jokonya, Thajasvarie Naicker, Takafira Mduluza.

**Writing – original draft:** Tariro L. Mduluza-Jokonya.

**Writing – review & editing:** Arthur Vengesai, Herald Midzi, Maritha Kasambala, Luxwell Jokonya, Thajasvarie Naicker, Takafira Mduluza.

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
