## [Decision Letter · Decision Letter 0]

7 Dec 2020

Dear Prof. Mduluza,

Thank you very much for submitting your manuscript "Algorithm for Diagnosis of Early Schistosoma haematobium using Prodromal Signs and Symptoms in Pre-school Age Children in an Endemic District in Zimbabwe" for consideration at PLOS Neglected Tropical Diseases. As with all papers reviewed by the journal, your manuscript was reviewed by members of the editorial board and by several independent reviewers. In light of the reviews (below this email), we would like to invite the resubmission of a significantly-revised version that takes into account the reviewers' comments. 

We cannot make any decision about publication until we have seen the revised manuscript and your response to the reviewers' comments. Your revised manuscript is also likely to be sent to reviewers for further evaluation.

Sincerely,

Gabriel Rinaldi

Associate Editor

Michael Hsieh

Deputy Editor

Reviewer's Responses to Questions

**Key Review Criteria Required for Acceptance?**

**Methods**

-Are the objectives of the study clearly articulated with a clear testable hypothesis stated?

-Is the study design appropriate to address the stated objectives?

-Is the population clearly described and appropriate for the hypothesis being tested?

-Is the sample size sufficient to ensure adequate power to address the hypothesis being tested?

-Were correct statistical analysis used to support conclusions?

-Are there concerns about ethical or regulatory requirements being met?

Reviewer #1: Although overall this is an interesting study, there are numerous omissions in the Methods section and parts that are difficult to follow, making it unclear what exactly the investigators did. The major omission is in the description of the statistical analysis of the results. Although the Results speak of a "CHARI Algorithm", this term isn't even mentioned in the Methods. Furthermore, exactly how this algorithm was formulated is unclear. Although the Results section indicates that parameters assessing risk of exposure were incorporated into the algorithm, this isn't described in the Statistical Methods subsection of the Methods. Additional comments:

1. The authors mention that a questionnaire was administered to parents/guardians of the children, but the content of this questionnaire isn't elucidated. It would be best if a brief description of the type of questions could be included, or even better, include the questionnaire as supplemental material. 

2. The composition of the ESI cohort is not readily apparent in the Methods. From the Results section, it appears that this cohort consists of children who were egg negative at baseline as well as at the first follow-up (4 weeks) but who were positive at the second follow-up at 16 weeks. It would be best if this were clearly defined in the Methods section so that there is no ambiguity.

3. Methods, line 134: the authors state that they included children at "high risk" but do not define how this was determined. Please clarify.

Reviewer #2: Comments presented in the attached document

Reviewer #3: All the above areas were met

**Results**

-Does the analysis presented match the analysis plan?

-Are the results clearly and completely presented?

-Are the figures (Tables, Images) of sufficient quality for clarity?

Reviewer #1: 1. Table 1 indicates that some children had history of previous anthelminthic treatment, but this was supposed to have been exclusionary. Please explain.

2. There are two different Table 3s but only one is referenced in the text. 

3. Lines 271-273: it would be useful to have a table showing the proportions of both ESI and non-ESI children with the different signs and symptoms, rather than just stating the proportions for the ESI participants. It would help visualize the comparison.

4. Table 3: as a footnote, please indicate what the AOR is in comparison to (i.e., children with no signs or symptoms).

Reviewer #2: I have reviewed this section and provide comments

Reviewer #3: All the above areas were met

**Conclusions**

-Are the conclusions supported by the data presented?

-Are the limitations of analysis clearly described?

-Do the authors discuss how these data can be helpful to advance our understanding of the topic under study?

-Is public health relevance addressed?

Reviewer #1: The main criticism of this paper is the conclusion that the algorithm could be used to identify children who could be treated early to prevent chronic schistosomiasis. However, the authors only passingly make reference to the stage-specific effectiveness of praziquantel, when in fact this could be a major barrier to using this algorithm to guide therapy in endemic areas. Praziquantel is effective against cercariae and very young schistosomula but not against older schistosomula, which is exactly the stage of development that the worms infecting the children identified by the algorithm are likely at. More discussion of this limitation is warranted.

Reviewer #2: Yes, see my other comments

Reviewer #3: Yes, the authors fulfilled the above

**Editorial and Data Presentation Modifications?**

Reviewer #1: 1. This paper could use a thorough editing to correct grammatical, verb conjugation and spelling errors as there are multiple throughout.

2. The term "infection" is preferred to "infestation" for schistosomiasis, since the latter is normally reserved for ectoparasites.

3. Abstract, lines 25-26: please state the timing of the follow-up visits and not just "for a period of 6 months".

4. Abstract, lines 43-44: please state the comparator (i.e., vs. non-ESI?).

5. Introduction, line 78: bladder carcinoma is a sequela of infection, not a marker of it.

6. Introduction, line 79: "may manifest" not "manifests".

7. Introduction, line 83: I would add the increased risk of HIV transmission to the list of potential sequelae of chronic genital lesions in sexually active women.

8. Introduction, line 86: I would specify that "parasitological diagnosis" refers to egg detection via microscopy.

9. Introduction, lines 110-111: this statement is somewhat misleading. Praziquantel is not very effective in treating "initial stages of infection" prior to development of adult worms. This statement should be qualified to acknowledge this.

10. Methods, line 125: "temperate" not "temperament".

11. Methods, line 146: please describe the TOrCHeS screen that was conducted. Were serological assays performed? If so, which ones?

12. Methods, line 147: why was a Widal test done as part of screening for this study and what is the relevance as an inclusion criterion?

13. Methods, line 181: I would just use the title, "microhematuria screen" rather than "urinary morbidity" since morbidity isn't really being evaluated.

14. Methods, line 202: should be "blood urea nitrogen" concentration, not just "urea".

15. Methods, line 205: please include the units for hemoglobin values.

16. Methods, line 209: which hepatitides were evaluated and which antigens/antibodies?

17. Methods, line 210: "Giemsa" not "geimsa".

18. Table 1: under the "S. haematobium negative" subheader there are two columns but it is unclear what the difference is between the two. 

19. Table 2: please add the time-points (e.g., 4 weeks, 16 weeks) to the columns. Also, it would be helpful to include mean egg counts rather than just the intensity category.

20. Table 3 (the second one): what does the "**" superscript refer to? It isn't defined.

21. Results, lines 294-299: this paragraph is extremely difficult to decipher. Please provide additional explanation and clarification as to what was actually done.

Reviewer #2: None

Reviewer #3: Minor revision

**Summary and General Comments**

Reviewer #1: The authors have presented an interesting study that presents novel findings on signs and symptoms associated with early S. hematobium infection. However, additional clarity regarding their methods needs to be added, as described above. Additionally, they need to address the significant limitation of treatment with praziquantel not being very effective prior to adult worm development, since this could significantly affect their conclusions.

Reviewer #2: The paper is worth for publication and address an important starting point in term of diagnosis and treatment of schistosomiasis in pre-school children

Reviewer #3: Minor revision

PLOS authors have the option to publish the peer review history of their article (what does this mean?). If published, this will include your full peer review and any attached files.

Reviewer #1: No

Reviewer #2: Yes: Humphrey D. Mazigo

Reviewer #3: Yes: Edford Sinkala
---

## [Editor Report · Decision Letter 1]

15 Apr 2021

Dear Prof. Mduluza,

Thank you very much for submitting your manuscript "Algorithm for Diagnosis of Early Schistosoma haematobium using Prodromal Signs and Symptoms in Pre-school Age Children in an Endemic District in Zimbabwe" for consideration at PLOS Neglected Tropical Diseases. As with all papers reviewed by the journal, your manuscript was reviewed by members of the editorial board and by several independent reviewers. In light of the reviews (below this email), we would like to invite the resubmission of a significantly-revised version that takes into account the reviewers' comments. 

The manuscript is being sent back to the authors as they have only provided answers to Reviewer 2. Comments raised by Reviewers 1 and 3 were not included in the Rebuttal letter submitted by the authors.

We cannot make any decision about publication until we have seen the revised manuscript and your response to the reviewers' comments. Your revised manuscript is also likely to be sent to reviewers for further evaluation.

Sincerely,

Gabriel Rinaldi

Associate Editor

Michael Hsieh

Deputy Editor

The manuscript is being sent back to the authors as they have only provided answers to Reviewer 2. Comments raised by Reviewers 1 and 3 were not included in the Rebuttal letter submitted by the authors.
---

## [Decision Letter · Decision Letter 2]

27 Jun 2021

Dear Prof. Mduluza,

We are pleased to inform you that your manuscript 'Algorithm for Diagnosis of Early Schistosoma haematobium using Prodromal Signs and Symptoms in Pre-school Age Children in an Endemic District in Zimbabwe' has been provisionally accepted for publication in PLOS Neglected Tropical Diseases.

Best regards,

Gabriel Rinaldi

Associate Editor

Michael Hsieh

Deputy Editor

Reviewer's Responses to Questions

**Key Review Criteria Required for Acceptance?**

**Methods**

-Are the objectives of the study clearly articulated with a clear testable hypothesis stated?

-Is the study design appropriate to address the stated objectives?

-Is the population clearly described and appropriate for the hypothesis being tested?

-Is the sample size sufficient to ensure adequate power to address the hypothesis being tested?

-Were correct statistical analysis used to support conclusions?

-Are there concerns about ethical or regulatory requirements being met?

Reviewer #1: After revision, these are now acceptable.

Reviewer #3: The issue raised has been attended to

**Results**

-Does the analysis presented match the analysis plan?

-Are the results clearly and completely presented?

-Are the figures (Tables, Images) of sufficient quality for clarity?

Reviewer #1: After revision, these are now acceptable.

Reviewer #3: No new issues raised

**Conclusions**

-Are the conclusions supported by the data presented?

-Are the limitations of analysis clearly described?

-Do the authors discuss how these data can be helpful to advance our understanding of the topic under study?

-Is public health relevance addressed?

Reviewer #1: After revision, these are now acceptable.

Reviewer #3: The conclusion is accepatble

**Editorial and Data Presentation Modifications?**

Reviewer #1: None.

Reviewer #3: Nil

**Summary and General Comments**

Reviewer #1: The authors have adequately responded to the critiques.

Reviewer #3: The paper is generally fine

PLOS authors have the option to publish the peer review history of their article (what does this mean?). If published, this will include your full peer review and any attached files.

Reviewer #1: No

Reviewer #3: **Yes: **Edford Sinkala

---

## [Editor Report · Acceptance letter]

28 Jul 2021

Dear Prof. Mduluza,

We are delighted to inform you that your manuscript, "Algorithm for Diagnosis of Early Schistosoma haematobium using Prodromal Signs and Symptoms in Pre-school Age Children in an Endemic District in Zimbabwe," has been formally accepted for publication in PLOS Neglected Tropical Diseases.

Best regards,

Shaden Kamhawi

co-Editor-in-Chief

Paul Brindley

co-Editor-in-Chief
